# Self-Organizing Intelligent Matter: A blueprint for an AI generating algorithm

## Abstract

We propose an artificial life framework aimed at facilitating the emergence of intelligent organisms. In this framework there is no explicit notion of an agent: instead there is an environment made of atomic elements. These elements contain neural operations and interact through exchanges of information and through physics-like rules contained in the environment. We discuss how an evolutionary process can lead to the emergence of different organisms made of many such atomic elements which can coexist and thrive in the environment. We discuss how this forms the basis of a general AI generating algorithm. We provide a simplified implementation of such system and discuss what advances need to be made to scale it up further.

## 1 Introduction

An AI generating algorithm (Clune, 2019) is a computational system that runs by itself without outside interventions and after a certain amount of time generates intelligence (though the general idea is much older than this reference). Evolution on earth is the only known successful system thus far that we know of. In this paper we propose a computational framework, and argue why it might constitute such general algorithm, while being computationally tractable on current or near future hardware.

Building such system successfully will take many iterations and require a number of advances. What we hope to provide however, is a general procedure, where better and better systems arise as a result of improving the elements of the system and of experimentation rather than a fundamentally new algorithm. As an example, we had such procedure for supervised learning since the 1980's - neural networks trained by back-propagation and stochastic gradient descent. To reach the current impressive performance, it required a number of clever improvements, such as rectified non-linearities, convolutions, batch normalization, attention, residual connections and better optimizers, but the overall algorithm hasn't changed.

### 1.1 Evolution

Evolution is the primary process by which our algorithm operates. We describe it here.

In machine learning, the word evolution is typically used to describe variations of the following process (Back, 1996). We have a number of individuals and an objective to optimize. We evaluate the individuals, select the ones with good values of the objective, and mutate them to produce the next generation. Over time, individuals that are better at optimizing the objective appear. The use of the word evolution in this context is perhaps unfortunate, as this process is quite unlike the evolution observed in nature (Stanley et al., 2017). The clearest difference we can see is in the outcome. The former results in a small variation of final individuals that are the best at the objective. The latter results in a coexistence of huge variation of individuals with different behaviors - it is open-ended.

Let us therefore discuss the basic operation of natural evolution. We have an environment built out of elements (atoms) that are organized into bigger units such as individual bacteria or animals or groups of these. Those classes of units that propagate (Joyce, 1994) into the future (e.g. replicate - we will discuss this in a moment) keep existing, while those that don't propagate cease to exist. There is no objective based on which units are selected for propagation. Different collections of units find different means of propagating.

An important mechanism for the coexistence of a large number of different solutions is niche construction (NicheConstruction, 2020). Different collections of individuals modify or form the environment for one another. For example a bacterium consumes a food and excretes waste products, which modifies the local environment, being either food or a toxic substance for other bacteria. Another example is a prey forming a food source for a predator. These systems are self balancing, for example too many predators means they can't find food and die off, and vice versa. This means that the coexistence of multiple solutions is present (and evolving). Collections of individuals in such systems have different means of propagating, rather then being selected by a global objective.

The lack of objective we believe, as argued in (Stanley & Lehman, 2015), is critical and fundamental to open-ended creation and coexistence of diversity and runs counter to most developments in machine learning. Conversely, a presence of an objective in a system likely leads to a collapse of diversity. To see that attempting to create an objective is problematic, let us therefore try to suggest one and see what the problems would be. One of the clearest objectives one might propose is to reward an individual for reproduction. However, a predator might then simply kill its offspring which would increase its chance of making another one. We could try to tweak or find other objectives, but this might lead to unwanted and unforeseen behaviours similar to the one described above. There are other issues. We don't actually make copies of ourselves, but instead mix genes from individuals (sexual reproduction), which we need to select. What do we actually want to reward? In addition, most of the time, propagation of species depends on individuals working together in a group, often sacrificing some members for the good of the group. What is then a real reproducing unit? This is the reason why the word "propagate" (Joyce, 1994) is more appropriate than "reproduce". What really happens is that those classes of groups of elements that are set "a certain way" propagate and those that are not, don't. Note that this does not contradict the evolution of an intrinsic reward - which is an evolved means of finding a good policy during the lifetime of a given individual.

Another example of evolutionary process is our society. People don't have children in proportion to some objective the society has set, or there isn't just one job or hobby that we all converge on and that is the "best". People engage in a large number of jobs and hobbies. At the same time, memes, values, work practices, company structures and many other emergent concepts propagate. All these things coexist, both cooperating and competing.

## 1.2 PRINCIPLES

The field of artificial life aims at producing life and an evolutionary process inside a computer (Langton, 1997; Ray, 1991; Lenski et al., 2003; Sims, 1994; Yaeger et al., 2011; Gras et al., 2009; Soros & Stanley, 2014), see (Aguilar et al., 2014) for a review. In seminal work on Tierra (Ray, 1991), Thomas Ray created an artificial life system in the substrate of assembly instructions in computer memory. The set of instructions is executed by a number of heads and one organism corresponds to one head. The system was initialized by a handcrafted sequence of instructions that when executed, will copy itself to another part of the memory. The executions undergo mutations. Some of these result in organisms that are unable to replicate, while some others get better at it. Some organisms find ways to use other organisms copying mechanism to copy, forming parasites. Then resistance to parasites evolves, and hyper-parasites, phenomenons of sociality and cheating are observed.

This process eventually peters out, and the quest ever since has been to create a system that is truly open-ended, where complexity keeps increasing without bounds (Standish, 2003). A number of principles that characterize and open-ended process has been proposed (Soros & Stanley, 2014; Taylor et al., 2016) Here we select two that that we find the most important (points 2 and 3) and introduce two new ones (points 1 and 4).

- There should be no built in notion of an individual and no built in operation for reproduction of an individual. Instead, these should be an emergent properties of collections of units, composing new collections of units or themselves.

- The evolution of new (here emergent) individuals should create novel opportunities for the survival of others (Soros & Stanley, 2014).

- The potential size and complexity of the individuals phenotypes should be (in principle) unbounded (Soros & Stanley, 2014).

- To *tractably* obtain intelligent agents, fundamental neural operations should be basic building blocks of the environment.

The first property is absent in majority of works on artificial life, except a few that we review later below. However, this property is quite close to being true in Tierra and Avida (Lenski et al., 2003). A given individual (organism), consisting of a sequence of instructions, actually has to construct a new one (create the new sequence of instructions). Then however, a new agent is declared, a new head is allocated for it, and there is a distinction in operations within and outside of/between the individuals. Properties two and three should really be consequences of property one, which we will see once we describe the system in section 2. Property four is introduced for tractability and again will become clearer below.

There are other approaches that studied diversity in evolution. To prevent collapse of diversity in objective based evolution, ideas such map elites (Mouret & Clune, 2015) or quality diversity (Pugh et al., 2016) have been proposed, that explicitly try to keep diverse set of solutions. These assume a small handcrafted space of variables that define what individuals are different. However, given such space, they show that searching for space of diverse solutions, leads to a better result on the final objective than optimizing the objective directly, as the search process is able to step through solutions that are not obviously directly on the path aimed at the objective. To remove an objective, (Soros & Stanley, 2014; Brant & Stanley, 2017) propose to give every organism a chance to reproduce but do so if it satisfies a minimal criterion, such as sufficient amount of energy. In the latter, they consider a set of mazes and their solvers. A maze is propagated to the next generation if there are solvers solving it and a solver is propagated if there are mazes it can solve. This results in a coexistence of many solutions in the population. The system however behaves somewhat like a random walk through solvable mazes and it would be good to find a system with a stronger selection pressure.

Another approach to create an open ended system is to co-evolve one agent that designs an environment (sets the layout of things and such) and another one that tries to solve it (Wang et al., 2020; Racaniere et al., 2019). The former maintains a collection of environment-agent pairs, and a new such pair is allocated if it is sufficiently far from the pairs in the collection according to a pre-specified objective that does not depend on the details of the environment (but on the ability of agents to solve the environment).

In the system that we propose that there is no special agent designing the environment - there is actually no concept of agent, and instead there is only an environment where agent-like organisms can emerge and reshape the environment from within. There neither is any explicit minimal criterion for an organism since there is no explicit operation of copy of individual, but such criterion will appear for any emergent individual.

## 2 PROPOSED SYSTEM

The real world is built out of elementary particles that interact and compose bigger entities. Our proposed environment (an aspiring AI generating algorithm) is as follows. The environment is built out of elements, but at much coarser scale. Each element contains a neural operation. This can be for example a matrix multiplication, an outer product, or more likely a sequence of such operators comprising a mini neural network. The elements interact with each other through some form of underlying rules, a type of physics, and through a direct communication of neural states.

There can be various implementations of this system. In section 3 we provide a grid-world implementation with basic elements lying on a grid, communicating with propagating signals or via an attention mechanism and with underlying physics implementing energy and chemical-like exchanges. Another example could be elements forming rigid pieces in a three dimensional space that can be attached using joints, that contain the neural operations, interacting through exchange of signals with nearby attached pieces and that set the torques on the joints. There could be several types of elements in the system, and not all need to have neural networks inside.

In section 3 we provide a grid world implementation that highlights a number of important properties. Along the way we discuss what advances need to be made to make this system powerful. However, the potential of the proposed system is unbounded and here are some of the features that this system supports. Larger units composed of several elements can be formed either by physical attachment (like a robot) or simply as a set of units that decide to communicate and form a whole.

There is no limit to the potential size of these units. These units can propagate in a number of ways - they can grow by taking over other elements in the environment (a colony), they can replicate by assembling new copies - moving appropriate collected elements to place (e.g. a robot assembling a copy itself from pieces), or self-assemble, or they can produce whole different units that either implement specialized functions (a useful machine) or units that are even better than their predecessors. The latter likely requires intelligence.

## 2.1 CAPACITY FOR INTELLIGENCE.

In this subsection we discuss why the computational system just proposed has the capacity to represent general intelligence. We provide two arguments. First, any neural algorithm in machine learning that we have created, and likely create in the future, can be written as a sequence of operations, such as additions, matrix multiplications, outer products and non-linearities, operating on states which are tensors. An example is the sequence resulting from the forward, backward, and optimizer operations of a neural network. Auto ML Zero (Real et al., 2020) realized this, and directly searched for the sequence of such operators along with connectivity to states on which they operate and was able to learn basic neural algorithms. Since these operators are fundamental building elements of our environment, and these elements can be made to communicate with arbitrary connectivity, all neural algorithms can be implemented in our system as well.

Second, the human brain is capable of general intelligence, and its computation closely resembles an online recurrent network (not trained by back-propagation in time). That is, it can be approximated by a function $F$ that updates neural and synaptic values online $h_t, W_t = F(h_{t-1}, W_{t-1}, x_t, v)$ where $h_i$ is a cell state of neuron $i$ and $W_{ij}$ is the state of the synapse connecting neurons $i$ and $j$, $x$ is an input and $v$ represent hyper-parameters such as connectivity, coefficients, and nonlinearities. To represent actual neural computations well enough (note that we are not interested in faithful representation but something with similar algorithm and computational power) we likely need to represent $h_i$ and $W_{ij}$ by small vectors, as suggested in (Gregor, 2020; Bertens & Lee, 2019). These computations are local and therefore to search for them, we need to search for relatively simple functions (with few hyperparameters). We can search for them automatically, as in Auto ML zero or other neural architecture search works (Zoph & Le, 2016; Pham et al., 2018; Liu et al., 2018; Elsken et al., 2018). The important point to remember from this, is that there likely exists an online neural update, of power at least as large as that of the human brain, on a system of a similar size in terms of computational capacity. We could implement the computation of a given element of the environment by a layer of such neurons. During the run of the environment, other elements can set the hyper-parameters $v$ as well as $h$, $W$ of a given element. Those groups that have elements with the right settings of these parameters, for example $v$'s that implement a good learning algorithm (the update of $W$), should enable a better propagation of this group compared to others.

## 2.2 AGENT HYPOTHESIS.

In our system there is no separation between an agent and an environment (AgentHypothesis, 2020) - there is only an environment. The elements themselves may or may not form units of evolution (Smith, 1986; Szathmáry et al., 2005) - entities that multiply, show heredity and where the heredity is not exact. In the former case, they could move autonomously, collect energy and replicate, and form bigger aggregates or replicating organisms because that provides an advantage. However, we propose to aim for the latter, where a certain minimal number of simpler cooperating units are need to propagate themselves.

There is a large amount of works that are related in one way or another to self-organization, and we can only review certain ones most relevant to our paper. The study of origins of life aims to understand how life evolved from non-living things. At some point in time in early earth, there were only simple combinations of elements: atoms combined into simple molecules, and at later time, there was the last universal common ancestor (Theobald, 2010) of all current life, which was already a rather complex organism containing core structures of bacteria. There are theories of how life emerged in the period in between, such as an auto-catalytic system enclosed in a membrane (Gánti, 2003) or a self-replicating molecule (Joyce, 2002). As a model of later, (Virgo et al., 2012) consider a physical system of shapes and succeed at finding length two chains that replicate. However, they

find that the shapes have to have a very specific features, arguing that such shapes are unlikely to emerge spontaneously from chemistry, and are therefore this process isn't a likely origin of life.

Simple substrates to study artificial life are cellular automata and related particle systems. (Gardner, 1970; Chan, 2020; Schmickl et al., 2016; Sayama, 2009; Ventrella, 2020; Nichele et al., 2017). In the former, elements are cells on a grid and in the later they have real valued coordinates and move. Their behavior is governed by simple rules that update their state based on their previous state and that of near neighbours. A goal is to find a set of rules that give rise to life, exhibiting replication, variation and heredity. This has proven quite difficult and while simple self replicating structures have been found, they don't satisfy these criteria of life. It is then especially difficult to imagine how to obtain an intelligent life from such simple rules in a tractable fashion. The proposal we put forward in this paper, is to use general neural networks of section 2.1 as the key parts of the elements, which can together form large neural networks, but with comparatively much fewer elements than basic cellular automata/particle systems. This is the reason for the word "intelligent" in the title of the paper. In addition, obtaining complex behavior should be much easier that tweaking cellular automata rules since we are starting with good learners - knowing the current capabilities of reinforcement learning agents, which we can use to jump-start the system.

Cellular automata can be implemented using convolutional neural networks (Wulff & Hertz, 1993; Gilpin, 2019). In study of morphogenesis (self-assembly of a given object/pattern), (Mordvintsev et al., 2020) trained a convolutional neural network that respects the structure of cellular automata to produce a given pattern starting with single cell or number of other patterns. Similar approach was pursued using compositional pattern producing networks (Nichele et al., 2017). These works however are not studying evolution.

Finally, there is a body of work on swarm systems and self-assembling robots. The former studies how can swarms of robots coordinate their behavior to accomplish tasks such as disaster relief or to study an emergent swarm behavior. The later, which is the closest in some ways to out proposal, studies how robots can self-assemble from smaller pieces. In (Weel et al., 2014), they consider a single type of (simulated) robot piece from which varying bodies are assembled through a process of evolution. There are some differences from what we propose: There is an explicit notion of an individual that is constructed in a central birthing facility according to an evolved template, an individual has a central brain (rather than composed one), and individuals replicate if they meet after a certain minimal distance from the birthing facility. While they do indeed aim for objective free evolution, the individuals do tend be selected based on how quickly they can get certain distance from the birthing area. Finally, in (Mathews et al., 2017) they use a physical cylindrical robots on a table, that can attach and detach to accomplish a task. When they attach, their controllers form one nervous system, where one chosen unit acts as a brain and the rest as the body. The controllers are implemented using a specific logic rather than neural networks. In our case we propose to use general learner neural networks, that can combine to form a bigger computational system (assembling a brain), that can in principle program newly added units (as we propose in section 3.1 below) and that undergo objective free evolution.

## 3  GRID VERSION OF SIM AND GENERAL PROPERTIES.

This section introduces an instance of the self-organizing intelligent matter (SIM). It also serves as a discussion of various points relating to SIM and gives more reasons of why we believe it can form an AI generating algorithm.

As discussed in subsection 2.2, there is no built-in notion of an agent and there really is just an environment. Seeing that, it feels unnatural to implement the system on two different platforms as is commonly the case - one for the physical part - such as a physics simulator, and one for the neural part - a neural network framework such as TensorFlow, PyTorch or Jax. Instead, we propose to make such system in a single platform. Because we would like to produce intelligent behavior, we need to run neural networks efficiently, and therefore the system needs to be implemented on one of the latter platforms. Because of its flexibility, we choose Jax.

Jax operates on tensors, which we use to store elements. The elements need to interact and have the ability to form flexible aggregates of arbitrary size (property 3, subsection 1.2). We first focus on

neural operations and describe the ideal system: the one we aspire for. We then talk about the steps we took towards implementing that system.

## 3.1 THE IDEAL SYSTEM

In our system, we place a neural network at every point of an $m \times m$ grid ($m$ up to 400 in our experiments), but in general, a different and flexible connectivity can be used. The ideal system would use the networks described in subsection 2.1 since, as we argued, they are likely able to compose general learners. At every point in time such network updates activations, weights and possibly hyperparameters $h_t, W_t, v_t = F(h_{t-1}, W_{t-1}, v_{t-1}, x_t)$. Each network outputs a number of "actions" that affect other elements - the cells of the grid. One fundamental action a network can take is to set the values of $h, W, v$ of neighbouring cells. This action can be executed if certain conditions are satisfied, such as the cell having enough energy - the energy concept and its updates in our system are described later (this is a minimal criterion for an element, not an emergent individual). What does such action allow? We give a few examples.

- The first example is to copy $v$ with a noise, set $W$'s and $h$'s to some random variables and keep $v$ constant through the lifetime of any cell. This corresponds to creating a new mini-brain (consisting of one cell) that has a very similar structure to the source and is untrained - in other words, a replication operation, but without replicating the learned content - similar to way children are created.

- The second example is to copy all the variables, with some noise. This creates copy with the same knowledge as the original cell.

- The third example is somewhere in between, transferring some knowledge and not other.

- The fourth example is to have a joint set of weights for an aggregate of cells, and select out those which are used in a given copy, analogous to cell differentiation in animal bodies or bee specialization in a colony of bees.

- The fifth example is setting the parameters (or some of them) to something quite different - programming them - that causes new aggregates of cells to perform useful functions for the original such as collecting energy - in effect creating useful machines for the original aggregate.

- In the final example, again, the new cells are programmed, but this time with the goal of creating new aggregates that are better than the original. The aggregates can be thought of as both organisms and machines - there is no distinction between the two in our environment. Here, machines create better machines by both designing better brains and better bodies. Such process requires intelligence, which is exactly what we are aiming for. We believe there will be a point in time in the evolution of the environment when this will happen, creating a self reinforcing loop of improving intelligence. Having neural operations as the fundamental elements of the environment is the reason that makes us believe that such state can be achieved in computationally tractable fashion in this type of system.

## 3.2 OUR PROPOSED STEPS TOWARDS THAT SYSTEM

Let us come back to describing the system we built. Since more powerful algorithms are not yet available, we use standard recurrent neural networks. We cannot use reinforcement learning to train these networks in a direct way - RL is an algorithm for optimizing an objective, which we fundamentally don't have here. We could try evolving such objective and such approach might be tractable. Instead we resort to what is usually done in these situations - simply evolve weights. We use the standard tanh recurrent network and the operation of setting the weight matrices and biases of new cell is simply copy with mutation. Alternative representation that is common are compositional pattern producing networks (Stanley, 2007).

The neurons, and all the other variables mentioned below (such as energy and chemicals), are placed on an $m \times m$ dimensional grid, with $m$ up to 400, Figure 1 comprising up to 160k networks. The second action that a network can output is to move to a neighbouring cell, which will swap the content of this cell with the new one (we experiment with allowing and disallowing this action).

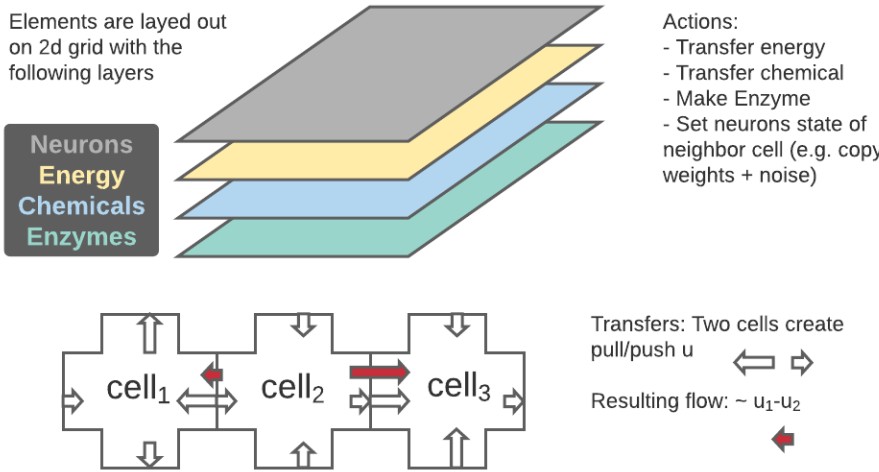

Figure 1: Diagram of the grid version of the system.

It is difficult to implement rigid bodies on a grid in Jax. To allow coordinated movements of larger units, we let the neurons communicate. For example they can communicate to all the cells belonging to a given group to move. One way to communicate is via local attention mechanism, where a given cell can read values of the $h$ of a cell in its neighborhood. This would also allow an aggregate of cells to implement a deep neural network, having different units representing different layers of a deep network. A single pass through such network takes a number of world updates. In our case we opt for a simpler communication mechanism, because a locally connected computation which local attention would utilize is not available in Jax. We create four signal layers that move in four directions (right, up, left, down). Each cell writes $h$ into all the layers, which then move and each cell can then read the content of all four signal layers.

Next we introduce a type of physics into the system, other than motion. We consider fields of energy and fields of $n_c$ types of chemicals $C_1, \ldots, C_{n_c}$ with $n_c$ typically $4 - 9$, at every location of the grid. A given cell can pull or push all these quantities in each of the four directions, which costs it energy proportional to the push or pull. Neighbouring cells can "fight" for these quantities, and if an energy of a cell falls below threshold, the cell is declared dead and its weights are set to zero. The cell also produces enzymes $Z_1, \ldots, Z_{n_c}$ that control the reactions in the respective order (e.g. $Z_2$ controlling $C_2 \rightarrow C_3$), that cost energy to make and that sum to at most one. A given reaction releases energy equal to $C_i Z_i$ except the last reaction which releases zero energy. As an example, to maximize energy, the cell should have one enzyme, say $Z_2$, pull chemical $C_2$ from outside (assuming some other cell is producing it), convert it to $C_3$ and excrete it to another cell. The cell also has a maximum lifetime and therefore it has to do something non-trivial (at least produce energy and copy) to propagate its information. If the population of live elements is below 10% we reawaken new ones with random weights. We also note that in our implementation, the elements themselves form autonomous units capable of self-replication, however, as discussed in section 2.2, in an ideal system they wouldn't.

There are two other variations that we experimented with. First we wanted to know how simple can one make the system and still observe and interesting behavior. We created a pure energy system, where instead of having chemicals, energy increases at every location up to some threshold. In the second system we experimented with different implementation. Instead of having a network at every point of the grid, we consider $n$ (1600) elements on $m \times m$ (200 × 200) grid that can move around. This time we used attention mechanism for communication within some distance between elements. We again only had a background energy, and the elements don't die or need anything for them to be "alive". They could just "sit around" (as atoms can). However, those that are active, can acquire energy by moving (energy gets automatically absorbed) and copy their weight onto others that have a lower energy. Those that do that will be the ones that propagate and overtake the system.

## 4 EXPERIMENTS

We run the system explained in the previous section. More detailed explanation and the settings of hyper-parameters are in the Appendix. The code will be released in a near future. What we observe is an exciting diversity among a series of runs, with snapshots shown in the Figure 2. This is best viewed in accompanying video [1]. It also shows the run of the system from the the start, showing competition among various classes of elements.

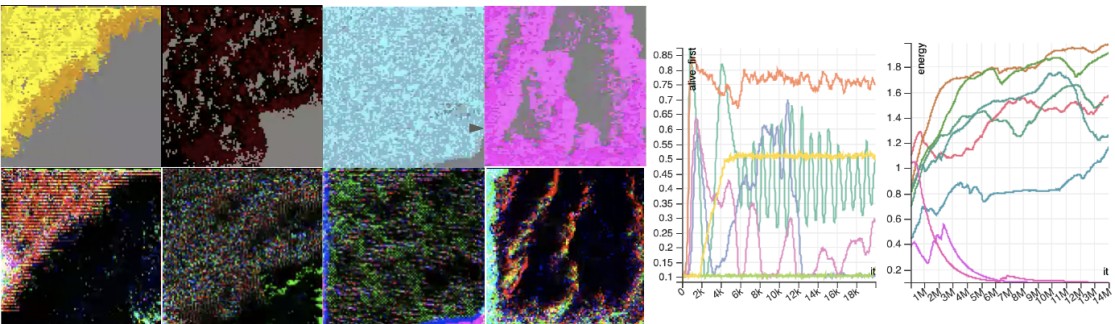

Figure 2: **Results of runs. Top row.** We selected three random weights and plotted them in RGB channels. This way elements with similar weights will have similar colour and those with different weights will likely have different colour. **Bottom row**. First three chemicals plotting in RGB channels. **Discussion.** We see a lot of diversity in the runs. These are best viewed in accompanying video. We often see coexistence of two species - seen in two different colours in the same region of space. We also see that the elements moved the chemicals around creating regions of high and low densities. In the top left square, we see that the region of low density is occupied by different species than the region of high density **First graph** Fraction of elements alive as a function of time from the start of the run (random new weights with no evolution in this run only). We see oscillatory behavior resulting from dynamics of two species. **Second graph** Energy content of elements in pure energy system of moving elements that need no energy to survive. However, those elements that collect energy can copy weights onto those with less energy, and those that do that propagate.

Looking closely at the Figure 2 top, we see in several regions a stable coexistence of two species of elements, represented by dots of different color in the same region of space. These persists for long amount of time, as can be seen from the videos, meaning they found a way to coexist, creating niches for one another.

Furthermore, looking at the Figure 2 top left, we see a light yellow region, containing two species, and an orange region containing different, third species. We see that the chemicals have been moved to sub-regions of space, and that the the species of the orange region live in a place with less chemical density. Thus the elements created niches by modifying density of chemicals in the environment. The challenge is not simply to be able to reproduce alone, but in the presence of others.

There are other types of behaviour we observe. In a smaller system, where we diffused the chemicals everywhere quickly, and turned off evolution for stability (weights are reset if density is less then 10%) we observed oscillations in population of two species, Figure 2 middle (and in the video) is reminiscent of the Lotka-Voltera system, (Lotka, 2002). The behavior in the pure energy grid system usually results in one species, but there are instances of two. This is similar for the version of neural elements not living in a grid. In Figure 2 right we see that in this system, despite not having any survival notion or an explicit selection for energy, the elements learn to collect energy, as this allows them to copy their weights onto others, which causes them to propagate.

---

[1]Videos: Best viewed by downloading (not viewing on the site): https://bit.ly/3nb6MCI (3.2Gb). Compressed version (more blurry): https://youtu.be/ifVjzhWL9ro

## 5 DISCUSSION

In this paper we proposed a framework for achieving intelligence by evolutionary process in an environment that is built out of interacting elements implementing computationally efficient and general learning. Only time will tell whether this framework is capable of such a feat. There are some critical advances that need to be made, most notably, creating general neural updates that can be evolved, or otherwise designed. Other directions for development are a way these elements interact, their embedding in a space and the underlying physics.

We have created a version of this system, that we believe has all the core necessary components, or would have if we had more powerful learning updates. It could form a starting point for the developments outlined in the previous paragraph. We summarize the core properties of this system that are general to all instances of SIM. There are elements containing neural networks. These elements can interact and communicate, forming larger units, in effect implementing larger networks. There is no objective based on which anything is selected. Instead, those compositions of units that find ways to propagate, will. There is no distinction between organisms and machines. The elements can write into other elements, they can do this by copying, or they can write other information to build machines that are useful for their creators or they can create whole new autonomous machines. The machines can create both new "bodies" - functional compositions utilizing physics, and new brains creating better algorithms than themselves.

A very intriguing question is what is the necessary physics needed for explosion of diversity and rise of intelligence. In the real world, the elements are elementary particles such as quarks and electrons. The former combine into protons and neutrons (there are few other particles such as photons), these combine into atoms. Few core atoms, primarily carbon, hydrogen, oxygen, nitrogen and phosphorus, form a wide diversity of molecules in the form of proteins and other types of molecules, eventually building cells as a basic units of life. What are the core rules that could give rise to complex life in the system we propose, where fundamental units themselves can already exhibit complex interaction and behavior? Does one need to introduce a complex chemistry or classical physics? An intriguing possibility, that we would like to explore in the future is whether any built in complexity is even necessary? What if we have the brain elements and only a basic rule - that of energy? Will natural selection, with such general learners, progressively construct more and more complex structures that outsmart one another? These are some of the exciting questions we plan to study in the future.

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

## A APPENDIX

### A.1 DETAILED DESCRIPTION OF THE SYSTEM.

We describe the system grid implementation in detail here. We consider a grid of sizes varying from $100 \times 100$ to $400 \times 400$. At every location of the grid we have the following variables: Energy (real scalar), chemicals ($n_c$ real scalars, typically 4, but up to 9), enzymes ($n_c$ real scalars). We also have signals carrying information about various variables as we describe below. Finally, we have recurrent neural network variables (we describe the network next): hidden layer activations of

size $n_h$ (typically 16), input to hidden weights, hidden to hidden weights, hidden biases, hidden to actions weights and action biases.

Next, we describe the neural network operations. At every point in time, except special times described below, we apply a classic recurrent neural network update to activations while keeping the weights fixed: $h_t = \tanh(W_x x + W_h h + b_h); \hat{a}_t = W_a h_t + b_a$, where $W$'s are weights, $h$ are activations and $\hat{a}$ are variables from which actions are computed (as described below). In the above formula we dropped the time index from $W$'s as they weren't updated at this time. The $W$'s were updated at the following times: If one cell (grid element) took a specific action (copy + mutate) aimed at neighbour cell, if it had sufficient energy (a threshold) and if the neighbour cell was empty - meaning having zero energy - the weights and biases got copied with added noise to the other cell.

As mentioned in the text, we use this update because we haven't yet discovered a good forms of general update. In sections 2.1, 3.1 we propose that such update would take the form $h_t, W_t, u_t = F(h_{t-1}, W_{t-1}, u_{t-1})$ where $u_t$ are the hyperparameters of the update rules. The simplest analogue to the previous paragraph would be that at every point in time except the same special times, $h, W$'s are updated (so the network can learn = update weights) while $u$'s are fixed. Finally, during the special times, $u$'s are not merely copied, but the network can generate new $u$'s in the neighbour cell, as well as $h$ and $W$, allowing the originator cell to program the new one.

Next let us describe the energy-chemical dynamics within the cell. Energy is a scalar that is bound between 0 and some max value. The chemicals transform in directions $Z_i \rightarrow Z_{(i+1) mod n_c}, i = 1, \ldots, n_c$ in proportion the amount of enzymes $C_i$ respectively and all the reactions release a fixed amount of energy except for $i = n_c$ which does not release any. The released energy is added to the energy of the cell. If energy falls below a threshold (by mechanisms described next), the weights and activations are erased - set to zero. We also use maximum lifetime of a cell after which the weights and activations are erased. The amount of enzyme as well as flows of quantities between the cells is controlled by the cell's actions which we describe next.

The cell has the following actions. 1. Copy action: from $\hat{a}$ we extract five components and take softmax to decide where and if to copy (0 - do not copy, 1-4 copy to one of the four directions). If a copy is successful, a fixed energy (tending to be large) is removed from the cell. 2. Move action: This is optional. If enabled, it has the same logic, but this time, the full content of the two cells (originator and target) is swapped. 3. Energy flow: The four values determine the proportion of energy the cells want to suck from each neighbour or push to the neighbour (positive vs negative). The cost of such action is proportional to the push. The resulting flow between two cells is obtained by subtracting the values of actions from the two cells, allowing two cells to fight or cooperate in moving the energy. 4. Chemical flow: The same logic but for each chemical. There are $4 n_c$ values, one for each chemical and direction. 5. Enzyme production: $n_c$ actions that create each enzyme. The total amount of enzyme is normalized to be at most one. There are two extra features. The energy is normally protected from falling below a threshold (protecting the cell from accidentally killing itself). The exception is that if the energy pull from the neighbour is sufficiently large, it will pull all the energy and kill the other cell (erase its weights and activations).

Finally there are signals. The cells need to know about what is happening at the other cells and to communicate to form larger aggregates and networks. We implemented the following communication. There are four information grids - one moving in each of the four directions. At every point in time, each grid moves in its direction. In addition, part of activations $h$ as well as energy, chemicals and enzymes overwrite to information grids for those cell for which energy is above threshold. The neural network input is formed by concatenating the four information grids at a given location.

Finally each run of the system was run on single GPU, and the largest system size $400 \times 400$ at $n_h = 16$ (160000 networks) was determined by the largest size we could fit into GPU memory.

