# OpenReview forum: "Self-Organizing Intelligent Matter:  A blueprint for an AI generating algorithm"
_ICLR.cc/2021/Conference — Reject_

### Official Review · AnonReviewer4 · 2020-10-19
**Stimulating topic, but description of scientific objectives, methodology and contributions should be improved**

**Rating:** 3
**Confidence:** 4

**Review:**


This paper discusses an artificial life approach to artificial intelligence, through 1) first discussing some general principles that may be useful in artificial life/chemistry systems from the perspective of open-endedness; 2) sketching an experimental system aiming to implement these principles and showing preliminary qualitative results. While the paper discusses a number of ideas that are highly relevant to AI and should be better known by the AI community, the paper has also some weaknesses and is not currently in a form that is ready for publication.

Strengths:
+ This paper's topic is a nice opportunity to enable the AI/ML community to know better about Alife approaches to artificial intelligence
+ The particular topic of open-endedness is also highly relevant, including in the perspective taken in this paper where one aims to model complex dynamical systems that do not assume a pre-defined concept of agent/individuals. This is useful in helping researchers in the AI community to deconstruct some of the assumptions of the models they are working on

Weaknesses:
- The paper does not explain clearly the relevance of the Alife approach to AI, in particular it does not explain well the historical roots of this idea that could enlighten AI readers (e.g. discussing foundational work such as Brooks and Steels, 1995), and does not analyze what fundamental contributions where made in the past, and what were the limits.
- Related to the point above, the paper does not position itself clearly in this vast literature: both the general/theoretical ideas and the experimental framework discussed overlap significantly with the large body of work in Alife. Clear statements of contributions would be useful.
- It is not clear what is the concept of "(general) intelligence" considered here, and whether there is a real value/utility in using this word, as opposed to saying the family of systems discussed in the paper is for studying the "self-organization and evolution of complex structures". Formulating the general problematics around theoretical questions on the origins and evolution of complexity would also enable better linking with the literature in the sciences of complexity and theoretical biology, which is not sufficiently made in the current version of the paper.
- It is not clear what is the logical rationale of the listing of principles considered in the paper: what is the scientific objective of this list of principles? how can one measure/evaluate their utility? where do they come from? Also, there seems to be some (at least apparent) inhomogeneity between them, e.g. point 1 says "there should be no built in notion of an individual", but point 2 says "the evolution of new emergent individuals create novel opportunities..." --> point 1 reads like an assumption while point 2 reads like a desirable emergent property.
- The proposed SIM system lacks formal definition (e.g. precise pseudo-code enabling precise understanding and reproducibility)
- While the paper aims for a general system with "general neural network controllers inside much fewer elements", in practice the system appears to have a lot of specific elements (e.g. physical/enzymatic systems) that are less generic than several recent work generalizing cellular automata as grids of neural networks with multiple kernels, e.g. Chan, 2020; Mordvintsev et al., 2020; Gilpin, 2019. Thus, it is not clear which properties are more general/differ/more useful than these other works
- There is no proposed methodology for evaluating the success/failure/progress of the approach: e.g. how could one characterize the self-organized structures that are qualitatively observed in the preliminary runs presented in the paper? What are precisely the scientific questions addressed by experiments, and how can experiments answer them?

I recommend the author(s) frame more precisely their work in terms of scientific objectives and links with the conceptual history of alife approaches to AI. The framing of the scientific objectives would benefit from a clear formulation of the scientific questions addressed, and a strategy for measuring progress towards answering them (in addition to the strategy for developing experimental tools presented in the paper, which requires more details to enable better understandability and reproducibility).

References:

Bert Wang-Chak Chan. Lenia and expanded universe. arXiv preprint arXiv:2005.03742, 2020.

Brooks, R. A., & Steels, L. (Eds.). (1995). The artificial life route to artificial intelligence: building embodied, situated agents. L. Erlbaum Associates.

William Gilpin. Cellular automata as convolutional neural networks. Physical Review E, 100
(3):032402, 2019.

Alexander Mordvintsev, Ettore Randazzo, Eyvind Niklasson, and Michael Levin. Growing neural cellular automata. Distill, 2020. doi: 10.23915/distill.00023. https://distill.pub/2020/growingca.

---

> ### Author Response · Authors · 2020-11-16
> **Response to review**
>
> Thank you for your extensive and insightful review. We agree on a lot of your points, and both discuss them below, and updated the paper: We positioned the paper better in literature (agent hypothesis section, and a reference in principles section, and we might still make some additions) and provided more complete description of the system (appendix). We feel that our paper presents a novel, consistent proposal with a number of insights and that could be a significant contribution for the path to AI and would be useful for ML community to know about (as you say yourself) so that others can contribute their ideas and works. It also provides a novel neural cellular automata system.
>
> Points 1, 2: We significantly expanded that section (now the limit is 9 pages starting with rebuttal phase) positioning the paper in the literature.
>
> Point 3: While studying “self-organization and evolution of complex structures” is actually really interesting, the primary purpose of this paper is to propose a framework for achieving general intelligence. While opinions and definitions on what this is vary, there is a general sense of what it means, such as “Ability to solve a large number of diverse problems in complex environments” or “being at least as good as humans in everything”.
>
> Point 4: Principles are useful summaries of the most important aspects of a problem. They guide design of algorithms and systems. For the other point: Points 2,3 mention individuals where we specifically put in the bracket “(here emergent)” meaning emergent individuals. We can emphasize that point more.
>
> Point 5 (formal definition). We modified the appendix to explain the system in detail.
>
> Point 6. The main purpose of the specific implementation in the paper is both for being able to better explain the proposed (abstract) system (for example explaining what would writing into another element look like) and to create elements of minimal implementation containing the core principles. However, as stated, we lack certain things to make this better, such as better neural operations, as well as more advances in enabling larger structures to form. Yet, it does form a novel neural cellular automata system.
>
> Point 7. We have indeed been trying to figure out how to make clear conclusions from these experiments, or from objective-free system in general. While we do show that it does evolves (e.g. system evolves to collects energy even if it doesn’t “have to”), more work on this needs to be done here. Again as mentioned - the issue is that the specific system is not as good as it should be - it both needs better neural algorithms and better way to form larger structures.
>
> We hope to continue building better such systems in the future both in terms of the “physics” rules and in terms of neural algorithms. However, this will take some time, and we felt that the overall framework and the demonstration has reached a sufficient level that it would be useful for other researchers to think about this and come up with different ideas and implementations, rather than waiting another year to make it work. Conferences are good venues for sharing such ideas.

---

> > ### Comment · AnonReviewer4 · 2020-11-20
> > **Follow-up discussion**
> >
> > Thanks for the answers!
> >
> > > We positioned the paper better in literature (agent hypothesis section, and a reference in principles section, and we might still make some additions)
> >
> > I appreciate the updates, but it is still unclear to me what are the main differences between the kind of system proposed and generalized cellular automata like those based on convolutional neural networks with hidden states or the Lenia Expanded Universe system. I think the existing systems are already very expressive. The authors say "These works however are not studying evolution": I think on the contrary these systems can be used as great tools to study evolution. They have indeed not presented a lot of experimental results in this perspective so far, but the present paper does not go much further either (the preliminary experimental results presented are not clearly different from many experiments on the emergence of dynamical patterns in cellular automata).
> >
> > > "and provided more complete description of the system (appendix).
> > We feel that our paper presents a novel, consistent proposal with a number of insights and that could be a significant contribution for the path to AI and would be useful for ML community to know about (as you say yourself) so that others can contribute their ideas and works. It also provides a novel neural cellular automata system."
> >
> > I agree with the aim, but to enable the community to build on this work, there needs to be a precise description of the new system proposed (even if it is just one instantiation). Some details were added in the appendix, but I think a precise pseudo-code with equations is needed, as well as access to an implementation of the corresponding code to enable reproducibility of the experiments.
> >
> > > Point 3: While studying “self-organization and evolution of complex structures” is actually really interesting, the primary purpose of this paper is to propose a framework for achieving general intelligence.
> >
> > My point was that if you aim for general intelligence, then I think you need to provide at least a coarse roadmap from the system you introduce to this general AI. Othewise, you could also just say we need to start from simulating Schrodinger's wave equations, and this will lead us to general AI as we've seen this was the case in our universe.
> >
> > Again, I'd like first to say that I think the approach is very interesting and could make great contributions to AI as well as would be suited for a conference like ICLR if updated with 1) a roadmap aiming to general AI; 2) a formal description + code of the proposed model(s); 3) more solid and systematic experimental results targeting more directly the evolutionary issues, associated with precise scientific questions. It seems the paper is not yet at this stage. However, in order to launch discussions with the AI community, preprints and workshops would seem well suited.

---

> > > ### Author Response · Authors · 2020-11-23
> > > **Follow-up discussion**
> > >
> > > We would like to claritfy:
> > > The statement "These works are not studying evolution" should have been "These works are not building artificial life systems" - and applies only the the last two works mentioned there, we will fix that.
> > >
> > > What those works do is to choose a problem - morphogenesis of a specific pattern say, and optimize the update rules of the cells in order to generate the pattern - either with bprop in neural network parameterized cellular updates for the first reference or with evolution in CPPN parameterized updates in the second reference.
> > >
> > > Lenia on the other hand is an artificial life system. There however the point is that it is hard to imagine how large scale neural networks needed for intelligence would both, evolve or, be efficiently implemented starting with such rules. Of the main points of the paper is to build these system out of such neural networks - and specifically ones discussed in the paper.
> > >
> > > Reponse to:
> > > "My point was that if you aim for general intelligence, then I think you need to provide at least a coarse roadmap from the system you introduce to this general AI. Othewise, you could also just say we need to start from simulating Schrodinger's wave equations, and this will lead us to general AI as we've seen this was the case in our universe."
> > >
> > > The description/framework we provide in the paper is a lot more specific description about how to build this than Schrodinger's equation and it is something that hasn't existed before.

---

### Official Review · AnonReviewer2 · 2020-10-26
**I am torn about this submission. On one hand, it discusses very important problems - the emergence of complex mathematical structures out of simple rules, on the other hand, it seems to lack substantial technical results to significantly contribute to these important problems.**

**Rating:** 4
**Confidence:** 3

**Review:**

The work follows the idea of artificial life on a computer and proposes to use RNNs as building blocks with simple rules of exchanging information between those individual RNNs on a grid.

The main weakness of this submission is the lack of the mathematical formulation of the algorithm that is used for generating the complexity. There are only two formulas in the manuscript: sections 2.1 and 3.1, which basically say that the authors consider a general dynamical system in discrete time with many degrees of freedom. This is not sufficient to explain the mathematical procedure used to arrive at figure 2.

There are many non-linear dynamical systems that can generate complexity (some of them are mentioned in the manuscript). Typically, the larger the number of degrees of freedom is, the more complex behaviors one obtains. The authors consider an extreme case when the overall dynamical system is composed of a collection of RNNs, with synaptic weights learned using some learning rule (unspecified in the manuscript). It is not surprising that a non-linear system with so many degrees of freedom can produce complex behaviors.

The authors present beautiful pictures of weights produced by their system. However, I do not understand if the authors are claiming that their dynamical system is somehow more meaningful than other more traditional counterparts. If so, I would appreciate a mathematical definition of the system considered, and a justification of this claim.  Or maybe the authors are saying that the precise mathematical form of the update rules for h and W is not important (and that’s why they don’t write them down explicitly in the manuscript) because they belong to some universality class, which often happens in studies of non-linear systems. Then, it seems necessary to provide evidence for this claim.

——————————

Thank you for your response. I have read the revised paper and the discussion with other reviewers. While I still think that the overall problematic discussed in this paper is important, I am not satisfied with the revisions. I think the paper would benefit from more technical content and clear mathematical definitions of the procedures used. I have updated my score to a reject.

---

> ### Author Response · Authors · 2020-11-16
> **Response to the review**
>
> Thank you for your helpful review. We hope this clarifies some of your concerns:
> * We understand that we had not provided enough details - we updated the appendix section with a more complete description of the system. We will also most likely release the code.
> * The desired update rules would be of the form h_t, W_t, u_t = F(h_{t-1}, W_{t-1}, u_{t-1}) where h’s are activations, W’s are weights and u’s are hyperparameters. W updates correspond to learning - in an online fashion. A given cell when writing to another one, would not only be able to copy u’s with mutation, but also write an arbitrary values of u’s as well as h’s and W’s. We (ML community) don’t have a good form of such updates - however we believe they exist as they correspond to updates that brains make. Because we don’t have them, we resorted to update (now explained in appendix) h_t = G(h_{t-1}, W, u) = tanh(W(x_t,h_{t-1})+b) where W’s and b’s are held fixed, except during mutations.
> * (response for two reviews) The primary purpose of the paper is to introduce a consistent framework for building an AI generating algorithm. One role of the grid implementation is to be able to explain and demonstrate the core parts of the system. It also provides a minimal implementation and a starting point from which a single framework (e.g. jax) implementation could be built, demonstrating a number of features one needs to think about. As we mentioned it misses certain core technologies, primarily better evolvable online networks that learn (rather than having weights fixed except at times of copy) and better mechanisms for forming larger aggregates. These need to be discovered in the future. However it does form a unique novel cellular automata system - see the expanded section “agent hypothesis” that positions the paper better in literature.

---

### Official Review · AnonReviewer1 · 2020-10-29
**Official Blind Review #1**

**Rating:** 5
**Confidence:** 4

**Review:**

This is an interesting paper that proposes a particular Alife framework to study the question of how to create an AI-generated algorithm. In the proposed system, each cell in a 2D grid-like environment is controlled by a different neural network. As far as I understand it, these neural networks are randomly mutated without rewarding the overall system for any particular objective. The authors study some of the dynamics that emerge from this framework.

While the main ideas in this paper are exciting (especially they idea that there is no direct concept of an agent),  it currently feels more like a workshop paper than a fully-fledged ICLR paper. Results are very preliminary, and while the emerging dynamics look interesting, the paper is missing a more systematic study of how different settings effect them. Additionally, I am slightly sceptical how these interesting patterns can lead to actual AI-generating algorithms. Without a task or some minimum criteria, what would drive the evolution of this system towards increasing complexity?

It would also be useful to further highlight the implications of this work for the larger ML and ICLR community. While certainly of interest to the Alife community, the potential importance of Alife domains for learning representations should be mentioned in more detail.

Additionally, some related work on neural cellular automates is missing:
- Growing Neural Cellular Automata, Mordvintsev et al.
- CA-NEAT: Evolved Compositional Pattern Producing Networks for Cellular Automata Morphogenesis and Replication. Nichele et al

[After rebuttal: Increased my score from 4 to 5. Still think the approach needs some more work to appeal to the broader ML community and in its current state would be best suited for a conference focused on Alife]

---

> ### Author Response · Authors · 2020-11-16
> **Response to review**
>
> Thank you for your helpful review. Let us address some of your concerns here.
> * (response for two reviews) The primary purpose of the paper is to introduce a consistent framework for building an AI generating algorithm. One role of the grid implementation is to be able to explain and demonstrate the core parts of the system. It also provides a minimal implementation and a starting point from which a single framework (e.g. jax) implementation could be built, demonstrating a number of features one needs to think about. As we mentioned it misses certain core technologies, primarily better evolvable online networks that learn (rather than having weights fixed except at times of copy) and better mechanisms for forming larger aggregates. These need to be discovered in the future. However it does form a unique novel cellular automata system - see the expanded section “agent hypothesis” that positions the paper better in literature.
> * Why does this paper makes sense for ICLR? Researchers at ICLR are interested in building AI (algorithms, data, environments). This is what this proposal is about: It is an unsupervised (evolution) method for creating both what would normally be considered an environment and for building the learning algorithm (through update rules for example).
> * References: We added them to the paper (the first paper is not directly about evolution, but really cool) and expanded the “Agent hypothesis” section, as well as provided more detailed explanation of the system in the appendix.

---

> > ### Comment · AnonReviewer1 · 2020-11-23
> > **Follow-up discussion**
> >
> > Thank you for the response.
> >
> > You wrote: "Similar approach was pursued using compositional pattern producing networks (Nichele et al., 2017). These works however are not studying evolution."
> >
> > What do you mean with they "are not studying evolution"? The paper by Nichele et al. also uses evolution to optimize neural networks.

---

> > > ### Author Response · Authors · 2020-11-23
> > > **Follow-up discussion**
> > >
> > > Ah, yes, that could be misleading. What we should have said is "These works are not building artificial life systems". We will fix that.
> > >
> > > What these works do is to choose a problem - morphogenesis of a specific pattern say, and optimize the update rules of the cells in order to generate the pattern - either with bprop in neural network parameterized cellular updates for the first reference or with evolution in CPPN parameterized updates in the second reference.
> > >
> > > In our case, we are building and ALife system, that has no objective to optimize, but instead, those behaviors which are able to expland/preserve/replicate well - will propagate into the future.

---

### Official Review · AnonReviewer3 · 2020-10-29
**Too explorative/especulative; the paper lacks some crucial references.**

**Rating:** 8
**Confidence:** 1

**Review:**

The paper describes a framework for artificial life, where basic building blocks are artificial neural networks (ANN) elements (matrix multiplication, and other linear algebra operations), intended to be open-ended and without any guiding objective. The authors emphasis that in their proposed approach there is no predefined distinction between environment and agents, as they attempt that the 'physics' of their proposed framework allow agents to emerge on their own.

Pros:
- The paper is very well written. Language is clear at times, the paper is well organized and the there is no overuse of math notation.
- The framework advanced by the authors seems solid enough to experiment on the emergence of life.
- Results need to be further analyzed, but they are encouraging at this stage of the research.
- Paper provides in-depth, or even philosophical, discussion relating topics such as emergence of life and current advances in computer science. It is very re-freshing to read a paper that is not only about surpassing other methods by a marginal score in benchmark datasets.

Cons:
- Although the paper provides a thorough discussion on related work, it is surprising that it does not make more emphasis on results related to cellular automata. Conway's Game of life seems to comply with their main aim at developing an environment where no distinction between agents and environment is made, and where agents emerge on their own. It would be good if authors provide a bit of contrast discussion on this issue.
- There are some strong claims across the paper or concepts being left without neither references or definitions. For example:
1. In Sec. 2.1 authors introduce the concept of "general intelligence". What exactly does that mean?
2. Also, in Sec. 2.1 authors claim that human brain computation closely resembles recurrent networks, but they do not provide a reference that backs such strong statement. Last time I check, there was little evidence that processes similar to backpropagation were happening in the brain, let a alone that artificial neurons models are now considered poor approximations of actual neurons.
3. The whole idea that life is objective-less can be controversial. A recent hypothesis [1] actually suggests that the life's objective is to accelerate entropy, that is, that emergence of life-like molecular structures, and even evolution itself,  are only a consequence, or even a particular case, of second law of thermodynamics. Authors should be more careful when dealing with such motifs.

References:
1. England, J. L. (2013). Statistical physics of self-replication. The Journal of chemical physics, 139(12), 09B623_1.

---

> ### Author Response · Authors · 2020-11-16
> **Response to review**
>
> Thank you for your helpful review and your appreciation of the paper. We modified the paper and address your comments here.
> * Cellular automata and game of life: We had a very brief discussion on cellular automata and particle systems (well, one sentence) which we significantly expanded in the current version of the paper (uploaded) - in the agent hypothesis section. Game of life is probably the original such system and we added the reference.
> * General intelligence: We just mean it in a loose sense, such as “Ability to solve a large number of diverse problems in complex environments” or “being at least as good as humans in everything” (including creating new things).
> * Brain as an RNN: We don't mean a classic RNN trained by back-propagation in time (the brain is not trained this way) - but in a looser sense: That the core computational/informational units are neurons with the states of their bodies (corresponding the activations) and the state of their synapses (corresponding to weights) as the main processing units, and that update in an online fashion as characterized in that paragraph. In this sense this is not controversial and we will clarify it in the text.
> * The idea that life is objective-less is controversial but the following is our opinion: For example if the purpose of life was to say accelerate entropy - how would it actually happen? There isn’t someone that rewards a given entity each time it increases an entropy so we think that such principle might be a consequence of evolution (and a useful characterization) rather than a purpose.

---

### Decision · Program_Chairs · 2021-01-07
**Final Decision**

**Decision:**

Reject

**Comment:**

This paper proposes a framework for artificial life. In the framework, there is no primitive agent construct, but rather a set of basic recurrent network components (such as linear algebra operations). The framework is open-ended and objective-less. The authors describe the emergence of different organisms out of these building blocks, illustrating the idea with a simplified implementation.

The paper is extremely original, at least in the context of the deep learning/ICLR community. At the same time, together with a majority of reviewers (mostly experts in relevant topics such as neuro-evolutionary methods and artificial life), I felt that this work is not ready to be presented at a major conference, for three main reasons. 1) It should make its links to the huge literature on artificial life outside the ICLR community clearer. 2) The empirical work is somewhat limited and 3) this is not counterbalanced by a clear theoretical roadmap.

I thus do not recommend acceptance, although I really hope the authors will present a more thoroughly worked out version of the paper soon.